# OpenReview forum: "Stop Before You Forget: NTK-Guided Early Stopping for Continual Learning"
_ICLR.cc/2026/Conference — ICLR 2026 Conference Withdrawn Submission_

### Official Review · Reviewer_oSLr · 2025-10-31

**Soundness:** 2
**Presentation:** 3
**Contribution:** 2
**Rating:** 2
**Confidence:** 3

**Summary:**

This paper proposes a proactive approach to continual learning, introducing the Forgetting–Acquisition Ratio (FAR) to detect gradient interference between new and past tasks. When FAR exceeds an adaptive threshold, training is stopped early to prevent catastrophic forgetting. Experiments on the GLUE benchmark report lower forgetting and modest accuracy gains compared to standard baselines.

**Strengths:**

1. The paper is well-structured and clearly presents both the intuition and the theoretical formulation.

2. The proposed Forgetting–Acquisition Ratio (FAR) is mathematically well-defined and theoretically grounded.

3. The integration with linearized LoRA makes the approach computationally efficient and easy to implement.

**Weaknesses:**

1. The paper does not clearly justify why NTK is an appropriate or insightful framework for modeling catastrophic forgetting in large, nonlinear models. The connection between NTK-space gradient projections and real forgetting dynamics in LLMs remains unjustified.

2. The approach assumes that a task's ``knowledge subspace'' can be captured by the span of gradients of datasets. This assumption is strong and untested. The paper provides no analysis or ablation to demonstrate that this subspace meaningfully represents task knowledge or that it remains stable under data variability.

3. The proposed mechanism of monitoring gradient conflict and limiting updates is conceptually close to many prior works, such as GEM (Lopez-Paz, D, 2017) and recent Zhao et al. (2023). However, the experiments do not compare against these directly related approaches, which weakens the novelty and makes it hard to assess the claimed advantage of the NTK-based formulation.


*Lopez-Paz, D. and Ranzato, M.A., 2017. Gradient episodic memory for continual learning. Advances in neural information processing systems, 30.*

*Zhao, Zhen, Zhizhong Zhang, Xin Tan, Jun Liu, Yanyun Qu, Yuan Xie, and Lizhuang Ma. "Rethinking gradient projection continual learning: Stability/plasticity feature space decoupling." In Proceedings of the IEEE/CVF conference on computer vision and pattern recognition, pp. 3718-3727. 2023.*

**Questions:**

1. Could the authors better justify why the NTK is an appropriate theoretical framework for modeling forgetting in large, nonlinear models such as LLMs? In particular, how do NTK-based gradient projections correspond to real training dynamics beyond the linearized regime? Any empirical evidence supporting this link would strengthen the claim.

2. Could the authors clarify the conceptual and practical differences between their NTK-based FAR method and prior approaches regarding gradient projections, such as GEM and A-GEM? Why were such methods not included in the experimental comparison?

---

### Official Review · Reviewer_8hu4 · 2025-10-31

**Soundness:** 1
**Presentation:** 1
**Contribution:** 1
**Rating:** 0
**Confidence:** 4

**Summary:**

This paper investigates the idea of using Neural Tangent Kernels (NTKs) as a criterion for early stopping during few-shot continual learning in order to prevent forgetting, specifically while training using Low Rank Adaptation (LoRAs). The authors propose a criterion based on the norm of the gradients and their projection to an orthogonal subspace of the gradient of previous tasks. They provide claims about the relationship between their metric and the loss and propose to use this criterion for early stopping. They perform experiments showcasing the effectiveness of their early stopping method.

**Strengths:**

- If the claims of the paper are true, it provides a novel and theoretically motivated method for early stopping during continual learning, which can potentially be used in future works to mitigate forgetting even without early stopping.

**Weaknesses:**

I see two **major** weaknesses with this work:
1. The experiments are very brief. Only the GLUE benchmark is evaluated, and the competing methods are relatively old (the latest being from 2019). More importantly, the authors do not compare to other LoRA continual learning works, such as [1], [2], which would be direct competitors to their method, and conceptually similar, as they involve orthogonal gradients/subspaces.

2. More importantly, perhaps, the paper suffers from **severe** issues regarding the theoretical claims. See questions below for a few that I could find and point out.


[1] Liang, Yan-Shuo, and Wu-Jun Li. "Inflora: Interference-free low-rank adaptation for continual learning." Proceedings of the IEEE/CVF Conference on Computer Vision and Pattern Recognition. 2024.

[2] Wang, Xiao, et al. "Orthogonal subspace learning for language model continual learning." arXiv preprint arXiv:2310.14152 (2023).

**Questions:**

I have many questions regarding the proofs of the theoretical results as provided in the appendix. Below are my questions/concerns with reference to line numbers, in ascending order.

1. Line 219 uses big $\tau$, but eq 8 uses small $\tau$. Same with Corollary 1.

2. Line 258: Considering Eq. 12, in Eq.13. $\\|g_B\\|$ cancels out, meaning the upper bound is proportional only to $\\|g_B^{\perp}\\|$, which is counterintuitive at best, and irrelevant at worst. This means the higher the norm of the **orthogonal** direction, the less we can guarantee a small forgetting. More on the proof of this theorem in the following points.


3. Line 895: Theorem 3: The function is said to be “smooth with parameter L”. I am assuming this refers to L-smoothness. L-smoothness is defined for functions mapping to $\mathbb{R}$, but the gradient is not such a function. Additionally, for the rest of the proof, it seems that L and $\mathcal{L}$ are used interchangeably, even though one is the loss and the other is the smoothness parameter.

4. Line 906: is said to be due to the Lipschitz Hessian condition. It is not mentioned that the Hessian is assumed to be Lipschitz, only that the gradient is L-smooth, which, as we established before, makes no sense. Let’s assume the authors meant that the Hessian is L-Lipschitz. Even then, I fail to see how the resulting integral has anything to do with that. At best, this is the Taylor approximation error, which I still fail to see, as it involves a double integral. The authors provide no citation of this derivation. Afterward, a new parameter $\rho$ is introduced, which is not defined or mentioned before. Finally, it is unclear how combining line 909 and 911 results in 914.

5. Line 952: Jacobian J used in Eq. A.4 is introduced in Eq. A.5 and not mentioned in the notation section of the appendix.


6. Line 964: “In the NTK regime, $\nabla LA(θ)$ lies approximately in $span(B_A)$”. I am assuming the authors mean the lazy regime? In any case, no citation or proof of this claim is provided.

7. Additionally, it is unclear how line 970 results in line 971. I can only guess it is an application of the Cauchy-Schwarz inequality, which does not explain how the norm of $g^{\\|}$ turns into the norm of $g^{\perp}$. Finally, the resulting theorem seems counterintuitive, as mentioned above about Theorem 1. Combined with the issues in the proof, I am quite skeptical of the validity of this result. If the authors can provide a convincing explanation, that would be great.

8. Line 972: “From the NTK theory, for gradients in the tangent kernel space.” Which NTK theory? Citation or proof is needed here.

9. Line 1033: “The Grassmannian distance is.” There is the Grassmannian, and the Grassmann distance. Not Grassmannian distance. Additionally, Line 1033 states the Grassmann distance is $d = \\|sin\Theta\\|_2$. This is incorrect; the Grassmann distance is defined by the square root of the sum of the squared canonical angles. If this is an additional result, the authors should provide a citation or proof.

---

### Official Review · Reviewer_x5LC · 2025-11-02

**Soundness:** 3
**Presentation:** 3
**Contribution:** 3
**Rating:** 4
**Confidence:** 4

**Summary:**

This paper tackles catastrophic forgetting in continual learning (CL) by detecting forgetting during training and stopping when it is predicted to occur. The method monitors the Forgetting–Acquisition Ratio (FAR), the fraction of the current update that is orthogonal to a past task’s subspace, to anticipate interference. The approach is evaluated on standard CL benchmarks and compared against several existing CL methods.

**Strengths:**

1. Rather than relying on traditional replay or regularization, the paper frames forgetting as overtraining and addresses it via schedule optimization (when to stop).
2. The procedure is straightforward and model-agnostic.
3. Results show that FAR correlates with actual forgetting and that the method improves both accuracy and forgetting reduction.

**Weaknesses:**

1. The idea is conceptually close to the early-stopping literature. The paper should articulate more clearly what is new and why it matters in CL.
2. The work would be stronger with theory linking FAR-based stopping to stability–plasticity trade-offs, and explaining why optimal stopping mitigates forgetting.
3. The schedule focuses on preventing forgetting, but gives less guidance on acquiring new knowledge, especially under similar / dissimilar / conflicting task relations. More analysis and strategies here would help.
4. The approach may not address shortcut (spurious) learning, where updates aligned with spurious features avoid the stop signal yet hurt generalization.
5. The comparison set would benefit from more recent CL baselines (2024–2025) to substantiate claims of improvement.

**Questions:**

1. What is the key novelty relative to traditional early stopping? In what ways does FAR provide information that validation loss or gradient-based heuristics do not?
2. How do you balance avoiding old-task forgetting with learning the new task? Have you analyzed the effect on forward transfer (not just retention)?
3. Can you provide theoretical intuition connecting your stopping dynamics to gradient interference or representation-drift accounts of forgetting?
4. How does the method behave under shortcut learning (spurious correlations)? Can FAR be adapted to detect or penalize shortcut-aligned directions?
5. How well does the approach generalize beyond the tested models (e.g., other PEFT methods, full fine-tuning, larger backbones)?
6. Can you include more recent baselines (published in 2024–2025) or explain why certain contemporary methods are not included?

---

### Official Review · Reviewer_iwMh · 2025-11-02

**Soundness:** 3
**Presentation:** 3
**Contribution:** 3
**Rating:** 6
**Confidence:** 3

**Summary:**

The paper addresses few-shot CL using LoRAs and proposes a proactive prevention strategy for CF based on NTK theory. Each prior task’s knowledge is characterized as a low-dimensional subspace obtained using SVD of LoRA parameter gradients. The authors propose 'FAR', the fraction of a new task’s gradient norm orthogonal to a previous task’s NTK subspace, to quantify interference across tasks. When FAR exceeds an adaptive threshold (based on task similarity), training is early-stopped to avoid irreversible forgetting. The method is integrated with linearized LoRA and claimed to add ~3% training overhead and no inference overhead.
The paper provides theoretical forgetting upper bounds using FAR (in the NTK linearized regime) and empirical results on GLUE benchmarks showing large reductions in forgetting and small memory and time overheads.

**Strengths:**

- The paper is generally well-written.
- The authors provide theoretical justification for the FAR-guided early-stopping under the stated assumptions.
- Empirical results on GLUE benchmarks show large reductions in forgetting with small memory and time overheads.

**Weaknesses:**

- The claim that all existing CF prevention strategies are reactive ("Most importantly, all these approaches have the same problem: they are reactive, attempting to mitigate forgetting after it has already occurred, rather than proactively preventing it.") is incorrect. A-GEM and several architecture-based CL approaches are not reactive.
- Key gradient-based baselines (GEM, A-GEM, OGD) are mentioned in related works but omitted from evaluations, even though FAR closely resembles these gradient-alignment approaches. Without such comparisons, it is difficult to assess how FAR with early stopping fares against projection-based methods.
- Experiments are limited to sequential GLUE tasks using RoBERTa-base + LoRA, which is a fairly narrow testbed. GLUE is small and standard, so the results don’t fully guarantee utility for general continual learning (e.g., for LLM adaptation). The effectiveness of FAR remains unclear under streaming domain shifts and cross-modal task sequences.

**Questions:**

- Early stopping may prevent learning of incoming tasks when there are large domain shifts. Is it possible to instead project or re-orient parameters instead of hard-stopping learning when the FAR threshold is exceeded?
- Is it compatible with tasks with variable numbers of classes? How are gradient matrices constructed in that case?

- (Minor) $\phi$ in equation 1 is not defined until later
- (Minor) typos: 'dimention' line 209, 'catastophic' line 253

---

### Note · Authors · 2025-11-12

I have read and agree with the venue's withdrawal policy on behalf of myself and my co-authors.